# Center of Mass Estimation During Single-Leg Standing Using a Force Platform and Inertial Sensors

**DOI:** 10.3390/s25030871

**Published:** 2025-01-31

**Authors:** Ryosuke Takahashi, Motomichi Sonobe

**Affiliations:** Department of Intelligent Mechanical Systems Engineering, Kochi University of Technology, Kami, Kochi 782-8502, Japan

**Keywords:** single-leg standing, force platform, inertial sensor, center of mass, double-inverted pendulum

## Abstract

Single-leg standing is a conventional balance evaluation method used in medicine. Although the center of mass (COM) displacement should be evaluated to determine balance quality, no practical COM estimation methods have been developed for single-leg standing. This study aimed to estimate the COM displacement in the anteroposterior and mediolateral directions during single-leg standing using practical measurements. We used a force platform and three inertial measurement units to estimate the COM displacement based on rigid-link models in the sagittal and frontal planes. The rigid-link models were composed of the stance leg, upper body, and non-stance leg. Seven healthy male subjects participated in the experiment to validate the estimation accuracy. The COM estimation accuracy was verified by comparison with measurements obtained using an optical motion capture system. The root mean square error of this method was 1.18 mm in the sagittal plane and 1.26 mm in the frontal plane. This technique will contribute to the detailed evaluation of individual balance abilities in the medical and sports fields.

## 1. Introduction

Single-leg standing is a more challenging task than bipedal standing [1] and has been used to evaluate Parkinson’s disease [2,3], anterior cruciate ligament reconstruction [4], fall risk [5], the impact of a history of sprain [6], and left–right asymmetry [7,8,9]. A subject is generally evaluated based on how long they can remain standing on a single leg [2,5,10]; however, the balance mechanism has not been discussed. To consider the balance mechanism in terms of control engineering, the relationship between the center of mass (COM) displacement and restoring force should be evaluated [11]. The restoring force is proportional to the difference between the center of pressure (COP) and the COM or COM acceleration [12].

A practical method for estimating the COM displacement during standing had not been established. Practical measurements should be performed without significant effort and space, but general measurement methods often require prior preparation, post-processing, and a measurement space. In contrast, an optical motion capture system using reflective markers, which is a typical COM displacement measurement method, is highly accurate [13]; however, it requires significant effort and a large space for measurement. Although this method enables a highly accurate COM estimation, it is unsuitable for clinical practice because of the effort required for measurement. Among other estimation methods, a method using a wearable inertial measurement unit (IMU) [14] and an integration method of horizontal acceleration obtained from a force platform [15,16] has been proposed. However, the estimation accuracy of these methods is considerably lower than methods that employ optical motion capture systems.

As a practical method, we proposed a COM estimation method for bipedal standing using a force platform and head IMU measurements [16]. This method uses the equations of motion of a double-inverted pendulum model composed of the foot, lower body, and upper body because this achieves an accuracy equivalent to that of an optical motion capture system. Three issues are associated with the application of this method to single-leg standing. First, because this method is limited to the double-inverted pendulum model, we must consider a model consisting only of the stance leg and upper body, excluding the non-stance leg. Second, during the COM estimation for the stance leg and upper body, the force received from the non-stance leg must be considered. Third, to obtain the whole-body COM, it is necessary to integrate the COM of the double-inverted pendulum with that of the non-stance leg obtained using another method.

This study aimed to estimate the COM displacement in the sagittal and frontal planes during single-leg standing using a force platform and three IMUs. This method uses a double-inverted pendulum model, which includes the stance leg and the upper body as the “main system” and applies the COM estimation method proposed in our previous study [16]. In addition, to consider the force acting on the main system from the non-stance leg, a “subsystem” consisting of the thigh and lower leg (including the foot) was defined. The motion of the subsystem was measured using two IMUs attached to the thigh and lower leg; the forces acting on the hip joint from the non-stance leg were estimated using an inverse dynamics analysis. Furthermore, the COM of the non-stance leg was estimated from the IMU attitudes of the thigh and lower leg. Consequently, the COMs of the main system and the subsystem were integrated to estimate the whole-body COM. The COM estimation accuracy of this method was verified by comparing it with the results obtained from optical motion capture measurements.

## 2. Estimation Methods

### 2.1. Mechanical Models

This section describes the mechanical model used for the COM estimation. Figure 1 shows the practical measurement system used in this study. The measurements included triaxial forces (*R_x_*, *R_y_*, *R_z_*) and moments of force (*N_x_*, *N_y_*, *N_z_*) obtained from a force platform under the stance foot, as well as triaxial acceleration vectors ${\tilde{a}}_{h},{\tilde{a}}_{s1},{\tilde{a}}_{s2}$ and angular velocity vectors ${\tilde{\omega }}_{h},\;{\tilde{\omega }}_{s1},\;{\tilde{\omega }}_{s2}$ obtained from the three IMUs attached to the thigh (*s*1) and lower leg (s2) in the non-stance leg and back of the head (*h*).

The human body was divided into a main system and a subsystem. Figure 2 shows the models of the main system ((a) and (b)) and the subsystem (c). The main system is a planar double-inverted pendulum model consisting of a foot fixed to the ground, a stance leg and an upper body, which can rotate around the ankle and hip joints. Point P represents the hip joint of the stance leg. We defined the mechanical models of the main system in the sagittal and frontal planes. The motion of the main system is represented by the stationary coordinate system *O*-*XYZ.* The *X*-axis is the horizontal forward direction, the *Y*-axis is the horizontal left-hand direction, and the *Z*-axis is the vertical upward direction. The subscripts *f*, *m*1, and *m*2 denote the feet, stance leg, and upper body, respectively.

The subsystem was defined in three-dimensional space as a double-link model with the thigh and lower leg (including the foot) of the non-stance leg. This system is connected to the main system at Q, which represents the hip joint of the non-stance leg. The motion of this system is represented by the translational coordinate system *Q*-*xyz*. The subscripts *s*1 and s2 represent the thigh and lower leg, respectively. The orientation of the translational coordinate system of the subsystem is the same as the orientation of the stationary coordinate system; however, their origins are different.

The physical parameters of each segment of the model are as follows: *m* is the mass, *L* is the segment length, *l* is the height of the segment COM from the bottom, *J* is the moment of inertia around the segment COM, and *w* is the length between the hip joints. These values were calculated as a function of the participants’ heights (*H* [m]) and weights (*M* [kg]), as shown in Table 1. The functions of mass, COM height, and moment of inertia were determined from the literature [17], and the function of segment length was determined from the literature [18].

The basic idea of COM estimation during single-leg standing was to decompose the system into a main system and a subsystem to estimate the COMs of each system. The COM estimation of the main system was based on the double-inverted pendulum model, which considered the force and moment of force acting from the subsystem. The COM of the subsystem was estimated from the rigid body attitude describing the thigh and lower leg of the non-stance leg. Finally, we estimated the whole-body COM by integrating both COMs.

### 2.2. Force and COM Estimation from Subsystem

This section describes the estimation methods for the COM displacement of the subsystem and the force and moment of force acting on the hip joint of the non-stance leg from the IMUs attached to the thigh and lower leg. The roll and pitch angles of the IMUs were estimated using an extended Kalman filter, and the yaw angles were obtained from angular velocity integration using the Euler method. The COM displacement of the subsystem was derived using a mechanical model based on the attitude angles of the two IMUs. The force and moment of force of the hip joint were derived via inverse dynamics.

The attitude angles of the two IMUs are expressed in terms of the 3-2-1 Euler angles *ψ_si_*, *θ_si_*, and *ϕ_si_* (*i* = 1,2). The angles *θ_si_* and *ϕ_si_* were estimated using an extended Kalman filter [19]. The yaw angle *ψ_si_* was estimated by integrating the angular velocity around the vertical axis because the accuracy of the geomagnetic information was insufficient. Using the *ψ_si_*, *θ_si_*, and *ϕ_si_*, the vector ${\tilde{r}}_{si}$ described in the sensor coordinate system can be transformed into the vector $r_{si}$ in the stationary coordinate system as follows:(1)$$\begin{array}{l} r_{si}=T_{si}{\tilde{r}}_{si}\\ T_{si}=\left[\begin{array}{ccc} \cos\theta_{si}\cos\psi_{si} & \sin\phi_{si}\sin\theta_{si}\cos\psi_{si}-\cos\phi_{si}\sin\psi_{si} & \cos\phi_{si}\sin\theta_{si}\cos\psi_{si}+\sin\phi_{si}\sin\psi_{si}\\ \cos\theta_{si}\sin\psi_{si} & \cos\phi_{si}\cos\psi_{si}+\sin\phi_{si}\sin\theta_{si}\sin\psi_{si} & -\sin\phi_{si}\cos\psi_{si}+\cos\phi_{si}\sin\theta_{si}\sin\psi_{si}\\ -\sin\theta_{si} & \sin\phi_{si}\cos\theta_{si} & \cos\phi_{si}\cos\theta_{si} \end{array}\right]. \end{array}$$

The COM displacements of the thigh ***r****_s_*_1_ = [*x_s_*_1_ *y_s_*_1_ *z_s_*_1_]*^T^* and lower leg ***r****_s_*_2_ = [*x_s_*_2_ *y_s_*_2_ *z_s_*_2_]*^T^* in the *Q*-*xyz* coordinate system can be expressed by the following equations:(2)$$\begin{array}{l} r_{s1}=T_{s1}\left[\begin{array}{c} 0\\ 0\\ -\left(L_{s1}-l_{s1}\right) \end{array}\right]\\ r_{s2}=T_{s1}\left[\begin{array}{c} 0\\ 0\\ -L_{s1} \end{array}\right]+T_{s2}\left[\begin{array}{c} 0\\ 0\\ -\left(L_{s2}-l_{s2}\right) \end{array}\right]. \end{array}$$

By integrating the COM displacements of the thigh and lower leg described in (2), we obtained the COM displacement of the subsystem ***r****_s_* = [*x_s_ y_s_ z_s_*]*^T^* as follows:(3)$$r_{s}=\frac{1}{m_{s1}+m_{s2}}\left(m_{s1}r_{s1}+m_{s2}r_{s2}\right).$$

The forces and moments of force acting on the hip joint of the non-stance leg were considered. The hip joint of the non-stance leg was assumed to be stationary. From the equations of motion of the thigh and lower leg, the force ***R****_s_* = [*R_sx_ R_sy_ R_sz_*]*^T^* and the moment of force ***N****_s_* = [*N_sx_ N_sy_ N_sz_*]*^T^* acting on the main system at the hip joint of the non-stance leg were derived as follows:(4)$$\begin{array}{l} R_{s}=R_{s1}+R_{s2},\\ R_{s1}=-m_{s1}T_{s1}{\tilde{a}}_{s1}+m_{s1}g,\;\;\;\;R_{s2}=-m_{s2}T_{s2}{\tilde{a}}_{s2}+m_{s2}g,\;\;\;g={\left[\begin{array}{ccc} 0 & 0 & -g \end{array}\right]}^{T}, \end{array}$$(5)$$\begin{array}{l} N_{s}=-N_{s1}-N_{s2}-r_{s1}\times R_{s1}-r_{s2}\times R_{s2},\\ N_{si}=T_{si}\left(\left[\begin{array}{ccc} J_{si,x} & 0 & 0\\ 0 & J_{si,y} & 0\\ 0 & 0 & J_{si,z} \end{array}\right]{\dot{\tilde{\omega }}}_{si}+{\tilde{\omega }}_{si}\times \left(\left[\begin{array}{ccc} J_{si,x} & 0 & 0\\ 0 & J_{si,y} & 0\\ 0 & 0 & J_{si,z} \end{array}\right]{\tilde{\omega }}_{si}\right)\right)\;\;\;(i=1,2), \end{array}$$
where ***R****_s_*_1_ and ***R****_s_*_2_ are the forces acting on the COMs of the thigh and lower leg, respectively; ***N_s_*_1_** and ***N_s_*_2_** are the moments of force around the COMs of the thigh and lower leg, respectively. These were derived from Euler’s equations of motion and provided as a function of the angular velocity and its numerical derivative.

### 2.3. COM Estimation

Here, we describe a method for estimating the COM displacement of the main system. In the sagittal plane, when the tilt angles of the stance leg and upper body are small, the linearized equations of motion shown in Figure 2a are obtained as follows:(6)$$m_{m1}{\ddot{X}}_{m1}+m_{m2}{\ddot{X}}_{m2}=-R_{x}+R_{sx},$$(7)$$\begin{array}{l} \left\{-\frac{J_{m1,y}}{l_{m1}}-m_{m1}\left(L_{f}+l_{m1}\right)+\frac{J_{m2,y}L_{m1}}{l_{m1}l_{m2}}\right\}{\ddot{X}}_{m1}+\left\{-\frac{J_{m2,y}}{l_{m2}}-m_{m2}\left(L_{f}+L_{m1}+l_{m2}\right)\right\}{\ddot{X}}_{m2}\\ \;\;\;\;\;\;\;\;\;\;\;\;\;\;\;\;\;\;\;\;\;\;\;\;\;\;\;\;\;\;\;\;\;\;\;\;\;\;\;\;\;\;\;\;\;+\left\{\left(m_{m1}+m_{m2}\right)g-R_{sz}\frac{L_{m1}}{l_{m}}\right\}X_{m}=N_{y}-N_{sy}-R_{sx}\left(L_{f}+L_{m1}\right), \end{array}$$
where *X_m_* is the COM displacement of the main system and ${\ddot{X}}_{m1}$ and ${\ddot{X}}_{m2}$ are the COM accelerations of the stance leg and upper body, respectively. In (7), we approximated (*L_m_*_1_/*l_m_*_1_)*X_m_*_1_ ≈ (*L_m_*_1_/*l_m_*)*X_m_*, because the COM displacement *X_m_*_1_ was an unknown variable. This assumes that the hip joint of the stance leg is in a straight line connecting the ankle joint and the COM of the main system. In addition, head acceleration is described as a function of the acceleration of the stance leg and upper body as follows:(8)$$\frac{L_{m1}\left(l_{m2}-L_{m2}\right)}{l_{m1}l_{m2}}{\ddot{X}}_{m1}+\frac{L_{m2}}{l_{m2}}{\ddot{X}}_{m2}={\ddot{X}}_{h}$$ where ${\ddot{X}}_{h}$ represents the horizontal acceleration of the head in a stationary coordinate system. From (6)–(8), the following simultaneous equations are obtained, where the unknown variables are ${\ddot{X}}_{m1}$, ${\ddot{X}}_{m2}$, and $X_{m}$ as follows:(9)$$\begin{array}{l} \left[\begin{array}{ccc} m_{m1} & m_{m2} & 0\\ A_{21} & A_{22} & A_{23}\\ \left(l_{m2}-L_{m2}\right)L_{m1}/l_{m1}l_{m2} & L_{m2}/l_{m2} & 0 \end{array}\right]\left[\begin{array}{c} {\ddot{X}}_{m1}\\ {\ddot{X}}_{m2}\\ X_{m} \end{array}\right]=\left[\begin{array}{c} -R_{x}+R_{sx}\\ N_{y}-N_{sy}-R_{sx}\left(L_{f}+L_{m1}\right)\\ {\ddot{X}}_{h} \end{array}\right],\\ A_{21}=-\frac{J_{m1,y}}{l_{m1}}-m_{m1}\left(L_{f}+l_{m1}\right)+\frac{J_{m2,y}L_{m1}}{l_{m1}l_{m2}},\;\;A_{22}=-\frac{J_{m2,y}}{l_{m2}}-m_{m2}\left(L_{f}+L_{m1}+l_{m2}\right),\\ A_{23}=\left(m_{m1}+m_{m2}\right)g-R_{sz}\frac{L_{m1}}{l_{m}}. \end{array}$$
By solving (9), we obtain the COM displacement of the main system and the COM acceleration of the stance leg and upper body.

The COM estimation method for the main system in the frontal plane is almost identical to that of the sagittal plane. When the tilt angles of the stance leg and upper body are small, the linearized equations of motion shown in Figure 2b are obtained as follows:(10)$$m_{m1}{\ddot{Y}}_{m1}+m_{2}{\ddot{Y}}_{m2}=-R_{y}+R_{sy}$$

(11)$$\begin{array}{l} \left\{\frac{J_{m1,x}}{l_{m1}}+m_{m1}\left(L_{f}+l_{m1}\right)-\frac{J_{m2,x}L_{m1}}{l_{m1}l_{m2}}\right\}{\ddot{Y}}_{m1}+\left\{\frac{J_{m2,x}}{l_{m2}}+m_{m2}\left(L_{f}+L_{m1}+l_{m2}\right)\right\}{\ddot{Y}}_{m2}\\ \;\;\;\;\;\;\;\;\;\;\;\;\;\;\;\;\;\;\;\;\;\;\;\;\;\;\;\;-\left\{\left(m_{m1}+m_{m2}\right)g-R_{sz}\frac{L_{m1}}{l_{m}}\right\}Y_{m}=N_{x}-N_{sx}+R_{sy}\left(L_{f}+L_{m1}\right)\pm R_{sz}w, \end{array}$$
where *Y_m_* is the COM displacement of the main system, and ${\ddot{Y}}_{m1}$ and ${\ddot{Y}}_{m2}$ are the COM accelerations of the stance leg and upper body, respectively. In (11), we approximated (*L_m_*_1_/*l_m_*_1_)*Y_m_*_1_ ≈ (*L_m_*_1_/*l_m_*)*Y_m_* because the COM displacement *Y_m_*_1_ was an unknown variable. The sign of the fourth term on the right-hand side of (11) depends on whether the stance leg is on the right or left. The sign is positive when the stance leg is on the left. The following head-acceleration conditions were included:(12)$$\frac{L_{m1}\left(l_{m2}-L_{m2}\right)}{l_{m1}l_{m2}}{\ddot{Y}}_{m1}+\frac{L_{m2}}{l_{m2}}{\ddot{Y}}_{m2}={\ddot{Y}}_{h},$$
where ${\ddot{Y}}_{h}$ denotes the horizontal acceleration of the head. From (10)–(12), the following simultaneous equations can be obtained:(13)$$\begin{array}{l} \left[\begin{array}{ccc} m_{m1} & m_{m2} & 0\\ B_{21} & B_{22} & B_{23}\\ \left(l_{m2}-L_{m2}\right)L_{m1}/l_{m1}l_{m2} & L_{m2}/l_{m2} & 0 \end{array}\right]\left[\begin{array}{c} {\ddot{Y}}_{m1}\\ {\ddot{Y}}_{m2}\\ Y_{m} \end{array}\right]=\left[\begin{array}{c} -R_{y}+R_{sy}\\ N_{x}-N_{sx}+R_{sy}\left(L_{f}+L_{m1}\right)\pm R_{sz}w\\ {\ddot{Y}}_{h} \end{array}\right],\\ B_{21}=\frac{J_{m1,x}}{l_{m1}}+m_{m1}\left(L_{f}+l_{m1}\right)-\frac{J_{m2,x}L_{m1}}{l_{m1}l_{m2}},\;\;\;B_{22}=\frac{J_{m2,x}}{l_{m2}}+m_{m2}\left(L_{f}+L_{m1}+l_{m2}\right),\\ B_{23}=-\left(m_{m1}+m_{m2}\right)g+R_{sz}\frac{L_{m1}}{l_{m}}. \end{array}$$

The COM displacement of the whole body (*X_b_*,*Y_b_*) is obtained by integrating the main system COM displacement and subsystem COM displacement as follows:(14)$$\left[\begin{array}{c} X_{b}\\ Y_{b} \end{array}\right]=\frac{1}{m_{m1}+m_{m2}+m_{s1}+m_{s2}}\left\{\left(m_{m1}+m_{m2}\right)\left[\begin{array}{c} X_{m}\\ Y_{m} \end{array}\right]+\left(m_{s1}+m_{s2}\right)\left[\begin{array}{c} x_{s}\\ y_{s}\pm w \end{array}\right]\right\}$$ where the sign of *w* is positive when the stance leg is on the right. The horizontal displacement of the hip joint during integration was ignored.

## 3. Experiments

### 3.1. Experimental Protocol

Seven healthy males (173.0 ± 2.9 cm, 60.6 ± 14.5 kg, 22.6 ± 0.9 years) participated in this experiment. Participants were selected based on their ability to stand stably on a single leg. This study was approved by the Ethical Review Committee of the Kochi University of Technology (approval number: 312). All the participants provided written informed consent. The participants stood on a single leg for 45 s on a force platform with bare feet. After practicing the preliminary experiment, the participants selected their preferred leg as the stance leg, as a result, all the subjects chose their right leg as their stance leg. The foot of the stance leg was placed at the center of the force platform. During the experiment, the subject stood with their knee on the stance leg extended, their hands on the hips, and their head facing forward. The non-stance leg was raised in front of the subject’s body. The experiment was repeated if the non-stance leg touched the ground or if the foot position of the stance leg moved significantly during the test.

To verify the influence of the motion of the non-stance leg on the accuracy of the COM estimation, the following three experiments were conducted, as follow: (A) normal single-leg standing, (B) repeated up-and-down movements of the non-stance leg, and (C) reciprocal rotational motion of the non-stance leg around the vertical axis. Experiment (B) verified the influence of the vertical force acting on the hip joint of the non-stance leg (*R_sz_*) for the COM estimation. This is because we approximated (*L_m_*_1_/*l_m_*_1_)*X_m_*_1_ ≈ (*L_m_*_1_/*l_m_*)*X_m_* in (7). Experiment (C) verified the influence of the drift error generated in the estimation of the attitude angle around the vertical axis of the two IMUs mounted on the non-stance leg by integration for the COM estimation. This is because we used the integral method instead of geomagnetic measurements. In (B) and (C), the participants moved their non-stance leg in response to the sound of a metronome at approximately 0.75 Hz. Each test was conducted three times for each participant.

### 3.2. Experimental Equipment

A force platform (TF-3040; Tec Gihan, Kyoto, Japan) and three IMUs (IMS-WD; Tec Gihan) were used in this experiment. These data were time-synchronized using measurement software (Ver. 5.1.1.0). The IMUs were attached to the occiput and outer surface of the thigh and lower leg of the non-stance leg. The positions of the IMUs were measured using a ruler to mount the IMUs close to the segmental COM height. An optical motion capture system (MAC3D System; Motion Analysis, Rohnert Park, CA, USA) was used to verify the accuracy of the proposed method. Twenty-nine reflective markers were attached to the subjects according to the Helen Hayes marker set to estimate the COM. The sampling frequencies of the force platform, IMUs, and optical motion capture system were set at 100 Hz.

### 3.3. Post-Processing

Two digital filters were used to process the measured signals. A zero-phase high-pass filter (pass frequency of 0.1 Hz) was applied to the horizontal forces *R_x_* and *R_y_* from the force platform to remove the drift because these variables were small. A zero-phase low-pass filter (pass frequency of 3 Hz) was applied to the estimated COM displacements and COP to remove any high-frequency noise.

An extended Kalman filter was applied to estimate the attitude of the IMUs. For the two IMUs attached to the non-stance leg, the yaw angle was estimated by integrating the angular velocity around the vertical axis. To suppress the integration drift, we assumed that the angular velocities around the vertical axes of the thigh (*Ω_s_*_1,*z*_) and lower leg (*Ω_s_*_2,*z*_) were equal. This integration was calculated using Euler’s method with an initial angle of zero. The accelerations obtained from each IMU were transformed into a stationary coordinate system using their respective attitude angles. We regarded these accelerations as the COM accelerations for each segment (head, thigh, and lower legs). The data obtained from the force platform and the IMU were synchronized with those obtained from the optical motion capture system based on head acceleration.

The estimation accuracy of the COM for the proposed method was evaluated using the Pearson’s correlation coefficient (*r*) and the root mean square error (RMSE) for the COM displacement obtained from the optical motion capture system as true values. The evaluation period was 30 s (from 10 s to 40 s). The correlation coefficient and RMSE are expressed by(15)$$r=\frac{\sum_{k=1001}^{4000}\left(q_{e}\left(k\right)-{\bar{q}}_{e}\right)\left(q_{t}\left(k\right)-{\bar{q}}_{t}\right)}{\sqrt{\sum_{k=1001}^{4000}{\left(q_{e}\left(k\right)-{\bar{q}}_{e}\right)}^{2}}\sqrt{\sum_{k=1001}^{4000}{\left(q_{t}\left(k\right)-{\bar{q}}_{t}\right)}^{2}}},$$(16)$$\mathrm{RMSE}=\sqrt{\frac{1}{3000}\sum_{k=1001}^{4000}{\left(q_{e}\left(k\right)-q_{t}\left(k\right)\right)}^{2}},$$
where *q_e_* is the estimated value obtained using the proposed method; *q_t_* is the true value obtained using the optical motion capture system; and $\bar{q}$*_e_* and $\bar{q}$*_t_* are the mean values of *q_e_* and *q_t_*, respectively.

## 4. Results

Figure 3 shows the typical COM estimation results in the sagittal and frontal planes from the proposed method (blue line) and the optical motion capture system (black line) for a representative subject (170 cm, 62 kg). The figure shows the results of the three types of experiments ((A), (B), and (C)). For reference, the COP displacements (red line) are plotted in the figures. Considering the COM obtained from the optical motion capture as the true value, the proposed COM displacement estimation showed considerably better agreement than the COP.

Table 2 and Table 3 show the mean and standard deviation of the correlation coefficient and RMSE for all the subjects. These results indicate that a highly accurate estimation of the COM displacement can be achieved in the normal single-leg standing test. The RMSE results showed that the accuracy of the sagittal plane estimation decreased in Experiment (B), in which the non-stance leg was moved vertically, and the accuracy of the sagittal plane estimation of the frontal plane decreased in Experiment (C), in which the non-stance leg was moved laterally.

## 5. Discussion

This study proposed a practical method for estimating the COM in single-leg standing using a force platform and IMUs and verified its accuracy by comparing it with the value obtained from an optical motion capture system. In normal single-leg standing (A), the ratio between the mean values of the RMSE and RMS (RMSE/RMS) for the COM displacement was 0.19 in the sagittal plane and 0.30 in the frontal plane, respectively. In a previous study describing the COM estimation method in bipedal standing from a force platform and a single IMU, the accuracy was 0.18 in the sagittal plane and 0.37 in the frontal plane [16]. The COM estimation method for single-leg standing had an accuracy similar to that of bipedal standing. The proposed method considers the force and moment acting on the hip joints from the non-stance leg estimated from two IMU measurements. Without considering the force and moment, the RMSEs of the COM displacement were 1.24 mm in the sagittal plane and 1.94 mm in the frontal plane in the normal single-leg standing experiment. These results indicate that consideration of the force acting on the hip joint of the non-stance leg is necessary to estimate the COM with a high accuracy.

Furthermore, we verified the estimation accuracy when the non-stance leg moved significantly. Because the swing leg moved vertically up and down in (B), the force acting on it affected the COM estimation in the sagittal plane. Subsequently, the RMS of the COM displacement increased slightly, whereas the RMSE increased significantly, particularly in the sagittal plane. The non-stance leg rotated horizontally around the vertical axis in (C), which affected the accuracy of the frontal plane estimation. This estimation error may be due to the error of the rotation angle estimation of the non-stance leg from the IMU measurements. However, even when the swing leg moved significantly, the calculated RMSE was approximately 2 mm. This estimation accuracy is sufficient for practical use because this value is approximately half of the RMS (approximately 5 mm), and even the representative example in Figure 3c (RMSE AP: 1.86 mm, ML: 1.57 mm) can sufficiently express the difference between the COP and COM.

In the proposed method, we considered the effect of the sampling frequency of the measurement device. The sampling frequencies of the force platform and IMUs were set to 100 Hz in the verification test. The COM displacement can be estimated from (9) and (13); however, because they are both algebraic equations, it can be estimated with the same accuracy regardless of the sampling frequency. However, the sampling frequency affects the performance of the filters when using a high-pass filter to remove the drift of the horizontal force, low-pass filter to smooth the estimated the COM displacement, and an extended Kalman filter to estimate the attitude of the head IMU. Although the sampling frequency is important for these performances, the appropriate frequency depends on the magnitude of the measurement noise of the instrument. In this experiment, 100 Hz was sufficient because the estimation performance did not change, even when the sampling frequency was increased.

The proposed method has several limitations. Because our validation test was limited to young men, the accuracy of the estimation methods for children, the elderly, and women were not sufficiently verified. This method uses physical parameters, such as mass, moment of inertia, COM position, and the length of each segment determined from the weight and height of a subject, which can cause estimation errors owing to differences in body shape. In this study, we assumed that the IMU was attached to the theoretical COM segment height. Because the IMU position affects the accuracy of the COM estimation, it should be able to estimate its own position [20]. In addition, the accuracy of the COM estimation decreased when the head moved independently of the trunk.

Some studies have discussed the balance differences between stable and unstable platforms during single-leg standing [21,22]. Theoretically, this COM estimation method can be applied to horizontally oscillating platforms. If this method is applied to control the support surface using real-time COM feedback, a balanced support system may be realized for single-leg standing.

## 6. Conclusions

In this study, we propose a practical COM estimation method for single-leg standing using a force platform and three IMUs. The estimation accuracy of the proposed method was almost the same as the accuracy of the estimation method used during bipedal standing (as reported in previous studies). To obtain the COM estimation accurately, the IMU should be preferably attached to the thigh and lower leg of the non-stance leg. This technology can enable the accurate evaluation of balance in the medical and sports fields.

## Figures and Tables

**Figure 1 sensors-25-00871-f001:**
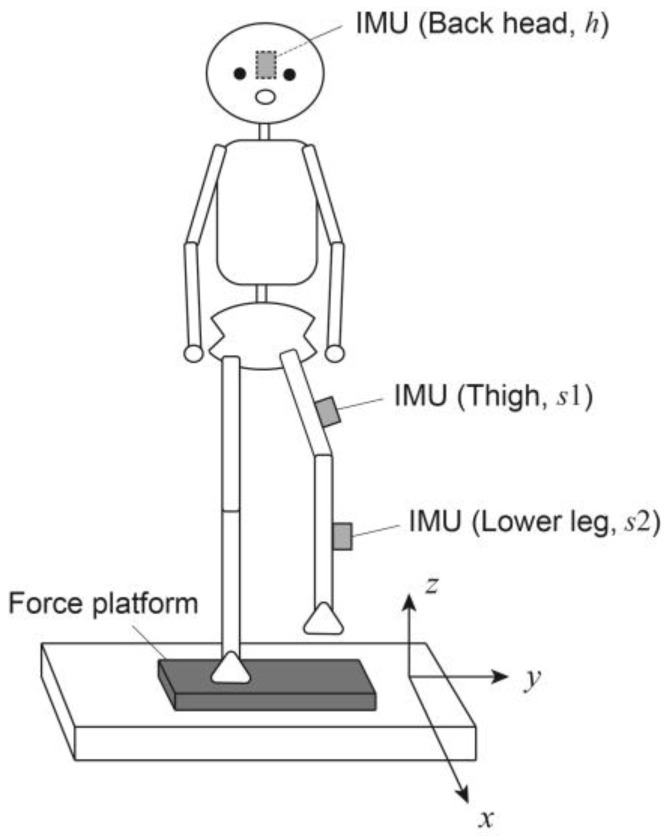
As a practical method, the measurement system consists of a force platform under the feet and three inertial measurement units (IMUs). The IMUs were attached to the back of the head, thigh, and lower leg of the non-stance leg.

**Figure 2 sensors-25-00871-f002:**
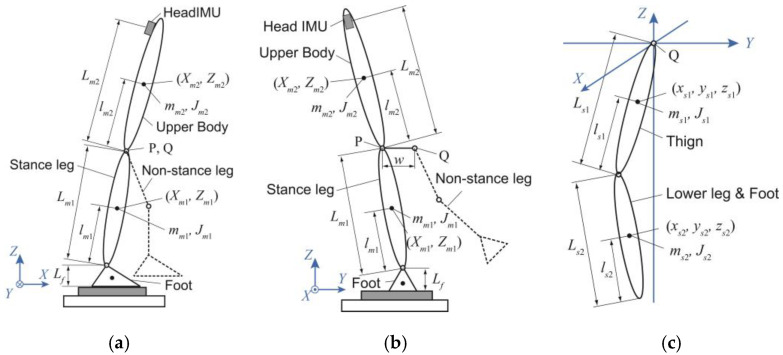
Rigid link models used in this study: (**a**) Two-dimensional mechanical model of the main system composed of the stance leg and upper body to estimate the center of mass (COM) in the sagittal plane. (**b**) Two-dimensional mechanical model of the main system for estimating the COM in the frontal plane. (**c**) Three-dimensional mechanical model of the subsystem describing the non-stance leg (thigh, lower leg, and foot) to estimate its COM and the forces acting on the hip joint. Point P represents the hip joint of the stance leg, and Q represents the hip joint of the non-stance leg. We assume that the points P and Q are always horizontal.

**Figure 3 sensors-25-00871-f003:**
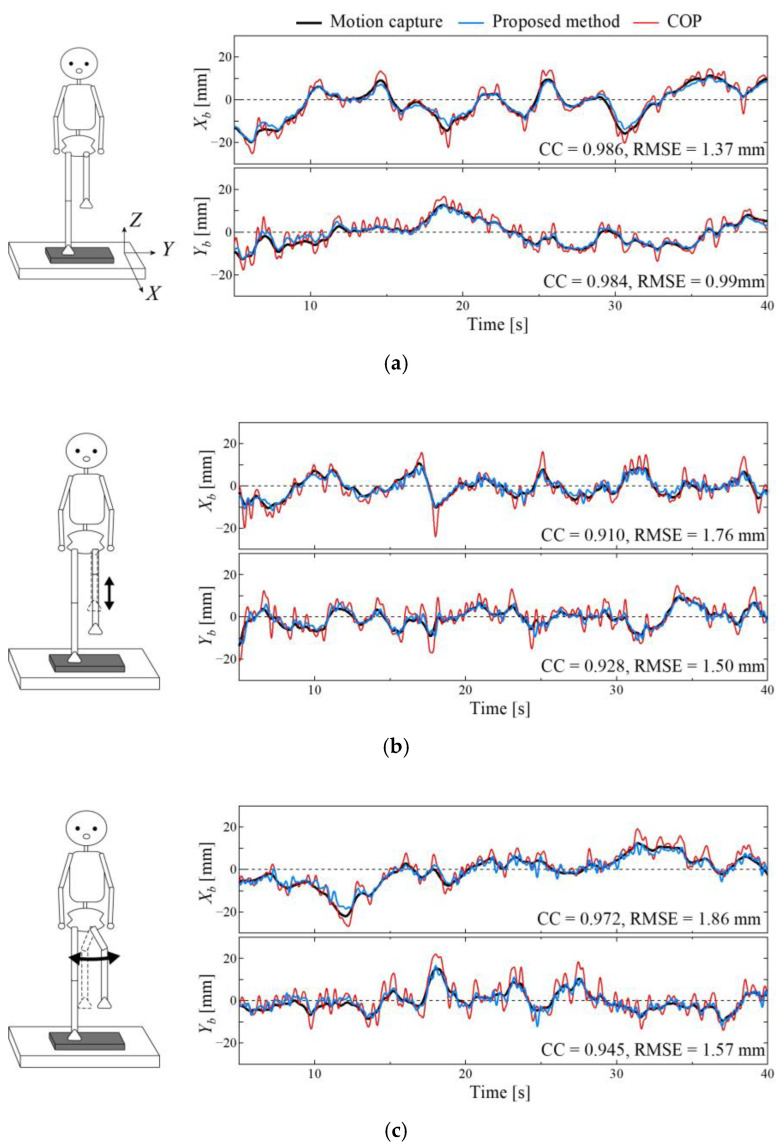
Time series data of the COM displacement for three motions of a subject (170 cm, 62 kg) in the sagittal and frontal planes: (**a**) Normal single-leg standing, (**b**) repeated up-and-down movements of the non-stance leg, and (**c**) reciprocal rotational motion of the non-stance leg. The blue line shows the COM displacement from the proposed method, the black line shows the COM displacement obtained from the optical motion capture results, and the red line shows the COP.

**Table 1 sensors-25-00871-t001:** Physical parameters of the mechanical model derived as a function of height and weight.

Segment	Symbol	Value	Segment	Symbol	Value
Upper body	*m_m_* _2_	0.656 *M*	Thigh	*m_s_* _1_	0.110 *M*
	*J_m_* _2, *x*_	1.30 × 10^−2^ *MH*^2^		*J_s_* _1, *x*_	4.81 × 10^−4^ *MH*^2^
	*J_m_* _2, *y*_	1.10 × 10^−2^ *MH*^2^		*J_s_* _1, *y*_	5.10 × 10^−4^ *MH*^2^
	*l_m_* _2_	0.169 *H*		*J_s_* _1, *z*_	1.53 × 10^−4^ *MH*^2^
	*L_m_* _2_	0.406 *H*		*l_s_* _1_	0.129 *H*
Stance leg	*m_m_* _1_	0.161 *M*		*L_s_* _1_	0.245 *H*
	*J_m_* _1, *x*_	2.53 × 10^−3^ *MH*^2^	Lower leg & Foot (Non-stance)	*m_s_* _2_	0.062 *M*
	*J_m_* _1, *y*_	2.56 × 10^−3^ *MH*^2^		*J_s_* _2, *x*_	4.86 × 10^−4^ *MH*^2^
	*l_m_* _1_	0.302 *H*		*J_s_* _2, *y*_	4.92 × 10^−4^ *MH*^2^
	*L_m_* _1_	0.491 *H*		*J_s_* _2, *z*_	2.89 × 10^−4^ *MH*^2^
Foot (Stance)	*m_f_*	0.011 *M*		*l_s_* _2_	0.155 *H*
	*L_f_*	0.039 *H*		*L_s_* _2_	0.285 *H*
Whole body	*l_m_*	0.589 *H*	*M*: Body mass [kg], *H*: Height [m]		
	*w*	0.100 *H*			

**Table 2 sensors-25-00871-t002:** Correlation coefficients (*r*) between the estimated values from the proposed method and those from the optical motion capture system for the COM displacement in the sagittal and frontal planes.

	(A)	(B)	(C)
Sagittal plane	0.985 ± 0.008	0.930 ± 0.051	0.920 ± 0.044
Frontal plane	0.951 ± 0.060	0.904 ± 0.081	0.877 ± 0.048

(AVG ± SD).

**Table 3 sensors-25-00871-t003:** Root mean square error (RMSE) between the estimated values from the proposed method and those from the optical motion capture system for the COM displacement in the sagittal and frontal planes. The root mean square (RMS) was calculated from the COM displacement obtained from the optical motion capture system.

		(A)	(B)	(C)
Sagittal plane	RMS	6.14 ± 1.43	6.28 ± 1.54	5.36 ± 1.16
[mm]	RMSE	1.18 ± 0.30	2.14 ± 0.64	2.06 ± 0.67
Frontal plane	RMS	4.21 ± 0.59	4.64 ± 0.71	4.56 ± 0.89
[mm]	RMSE	1.26 ± 0.69	1.99 ± 0.73	2.30 ± 0.62

(AVG ± SD).

## Data Availability

DOI: 10.6084/m9.figshare.28091513. Description: The COM displacements obtained from the optical motion capture system and the proposed method were saved as CSV files. The correlation coefficients and root mean square errors between the motion capture and the proposed method were also stored.

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
