# Peer review of "Center of Mass Estimation During Single-Leg Standing Using a Force Platform and Inertial Sensors"

_sensors, 2025, doi:10.3390/s25030871_

Round 1

Reviewer 1 Report

Comments and Suggestions for Authors

This paper focuses on COM estimation during single-leg standing. I believe the theoretical foundation is solid, and the results appear to be promising.

One minor point for consideration is the use of an IMU attached to the back of the head. The authors seem to assume that the head and upper body share the same attitude. However, there could be a significant discrepancy between these attitudes, as the head may move independently of a relatively stationary upper body. Does the lack of upper body attitude measurement impact the accuracy of COM estimation?

Author Response

Thank you for your review. I respond to your comments as follows.

Comments 1: One minor point for consideration is the use of an IMU attached to the back of the head. The authors seem to assume that the head and upper body share the same attitude. However, there could be a significant discrepancy between these attitudes, as the head may move independently of a relatively stationary upper body. Does the lack of upper body attitude measurement impact the accuracy of COM estimation?

Response 1: Thank you for pointing this out to us. As you pointed out, the head can move independently of the torso, and in such cases, the accuracy of COM displacement estimation using the proposed method will decrease. We have added this description to the limitations of the fourth paragraph in Discussion. On the other hand, the weighting of vestibular function increases in single-leg standing, so humans need to suppress the horizontal acceleration of the head. When the trunk and head move independently, maintaining single-leg standing becomes difficult.

Reviewer 2 Report

Comments and Suggestions for Authors

Dear authors,

I have reviewed your manuscript titled "Center of Mass Estimation During Single-leg Standing Using a Force Platform and Inertial Sensors." The study presents an interesting approach to estimating the center of mass (COM) displacement during single-leg standing, which has potential applications in medical and sports fields. However, I have some suggestions and minor changes that I believe will improve the clarity and rigor of your manuscript.

Introduction and Methodology:

L33: The term "practical method" is used frequently, but it is not well-defined. Force plates are expensive and require laboratory settings, making them less practical for widespread use. IMUs are more practical, but they alone cannot obtain COM. I suggest removing the term "practical method" or clarifying what makes your method practical compared to others.

L43: You mention "head IMU measurements" but do not specify the IMU at the leg. Please include this detail for clarity.

L46-52: This section seems more appropriate for the discussion rather than the introduction. Consider moving it to the discussion section.

L54-55: This information is repeated and can be consolidated or removed.

L56: When referring to a "previous study," please cite the specific study for clarity.

L68-70: This information is repeated and can be consolidated or removed.

Tables and Figures:

Table 1: Please change "Body weight" to "Body mass" for consistency with scientific terminology.

Experimental Protocol:

L164: The height measurement should be reported as 173.0 ± 2.9 cm for consistency in reporting significant figures.

L165: Please include the inclusion and exclusion criteria for the participants. Also, confirm that all participants signed an informed consent form.

L166: Did you verify that the right leg was the dominant leg for all participants? If not, please clarify how the stance leg was chosen.

Data Collection and Analysis:

L195: Address the issue of sampling at 100 Hz. Justify why this sampling rate is sufficient for your study.

L213: The Pearson's correlation coefficient is traditionally denoted as 'r.' Please use 'r' instead of 'CC' for consistency with standard notation.

Table 2: Specify that the number after ± is the standard deviation. Add this information to the table caption for clarity.

Discussion:

L257: The claim "This estimation accuracy is sufficient for practical applications" needs justification. Please provide a rationale or evidence to support this statement.

General Suggestions:

Ensure that all acronyms are defined at their first use.

Check for consistency in the use of units and significant figures throughout the manuscript.

Consider adding a limitations section to discuss the potential drawbacks of your method and how they might be addressed in future studies.

I believe these changes will enhance the clarity and rigor of your manuscript. I look forward to seeing the revised version.

Author Response

Thank you for your suggestion. We have revised the manuscript based on your feedback. The following are our responses to your comments.

Comments 1:

L33: The term "practical method" is used frequently, but it is not well-defined. Force plates are expensive and require laboratory settings, making them less practical for widespread use. IMUs are more practical, but they alone cannot obtain COM. I suggest removing the term "practical method" or clarifying what makes your method practical compared to others.

Response 1:

Thank you for pointing this out. Since there was a lack of explanation regarding practical measurements, we have added the description to the second paragraph of the Introduction.

Comments 2:

L43: You mention "head IMU measurements" but do not specify the IMU at the leg. Please include this detail for clarity.

Response 2:

This paragraph outlines our previous research [16] and presents the issues encountered when applying it to single-leg standing. However, because our writing was incorrect and the reference citation numbers were missing, we gave you a misunderstanding. We have corrected “propose” to the past tense and added a reference number.

Comments 3:

L46-52: This section seems more appropriate for the discussion rather than the introduction. Consider moving it to the discussion section.

Response 3:

Thank you for your comments. However, we suspect that this comment was probably due to a mistake in our expression, as mentioned above. In this paragraph, we clarify the issues encountered when applying the bipedal standing method to single-leg standing and explain the novelty of this paper. We believe that the description in the Introduction is appropriate.

Comments 4:

L54-55: This information is repeated and can be consolidated or removed.

Response 4:

Thank you for your comments. We suppose that your point is based on a misunderstanding of the third paragraph of the Introduction, as well as Comment 3. Because this paragraph is an explanation of the method proposed in this paper, I believe it is necessary to explain it, although there are some overlapping parts.

Comments 5:

L56: When referring to a "previous study," please cite the specific study for clarity.

Response 5:

Thank you for this suggestion. We have added a reference [16] to this section.

Comments 6:

L68-70: This information is repeated and can be consolidated or removed.

Response 6:

Thank you for pointing this out. For simplification, we have deleted the first sentence of Section 2.1 and replaced it with an overview sentence.

Comments 7:

Table 1: Please change "Body weight" to "Body mass" for consistency with scientific terminology.

Response 7:

Thank you for your comments. I have corrected it to “body mass” as you pointed out.

Comments 8:

L164: The height measurement should be reported as 173.0 ± 2.9 cm for consistency in reporting significant figures.

Response 8:

Thank you for pointing this out. We have corrected it.

Comments 9:

L165: Please include the inclusion and exclusion criteria for the participants. Also, confirm that all participants signed an informed consent form.

Response 9:

Thank you for this suggestion. Participants were required to be able to stand stably on a single leg. No other exclusion criteria were applied. We have added a description of this in section 3.1. We have also added a statement regarding informed consent to this section.

Comments 10:

L166: Did you verify that the right leg was the dominant leg for all participants? If not, please clarify how the stance leg was chosen.

Response 10:

Thank you for your confirmation, and we have not confirmed the dominant leg. Before the experiment, participants practiced standing on a single leg and selected their preferred leg. These descriptions have been added to section 3.1.

Comments 11:

L195: Address the issue of sampling at 100 Hz. Justify why this sampling rate is sufficient for your study.

Response 11:

Thank you for pointing this out. In our experimental devices, even if the sampling frequency was set to 100 Hz or higher, the accuracy of COM displacement estimation did not improve. We have added an explanation of the relationship between sampling frequency and estimation accuracy to the third paragraph of the Discussion section.

Comments 12:

L213: The Pearson's correlation coefficient is traditionally denoted as 'r.' Please use 'r' instead of 'CC' for consistency with standard notation.

Response 12:

Thank you for pointing this out. The CC in this article has been corrected to r.

Comments 13:

Table 2: Specify that the number after ± is the standard deviation. Add this information to the table caption for clarity.

Response 13:

Thank you for pointing this out. The description of AVG ±SD has been added to Tables 2 and 3.

Comments 14:

L257: The claim "This estimation accuracy is sufficient for practical applications" needs justification. Please provide a rationale or evidence to support this statement.

Response 14: Thank you for pointing this out. We have added a reason for judging it as sufficiently accurate at the end of the second paragraph of the Discussion section.

Comment 15:

Ensure that all acronyms are defined at their first use.

Response 15:

Thank you for your comments. We confirmed that there were no problems with the abbreviations (COM, COP, IMU, and RMSE) used in this paper. The explanation of RMS is missing in Table 3; therefore, we have added it.

Comments 15:

Check for consistency in the use of units and significant figures throughout the manuscript.

Response 15:

We have confirmed that there are no problems.

Comments 16:

Consider adding a limitations section to discuss the potential drawbacks of your method and how they might be addressed in future studies.

Response 16:

The limitations of this study are described in the fourth paragrap